# Fast Active Set Methods for
# Online Spike Inference from Calcium Imaging

**Johannes Friedrich**[1,2]**, Liam Paninski**[1]
[1]Grossman Center and Department of Statistics, Columbia University, New York, NY
[2]Janelia Research Campus, Ashburn, VA
j.friedrich@columbia.edu, liam@stat.columbia.edu

## Abstract

Fluorescent calcium indicators are a popular means for observing the spiking activity of large neuronal populations. Unfortunately, extracting the spike train of each neuron from raw fluorescence calcium imaging data is a nontrivial problem. We present a fast online active set method to solve this sparse nonnegative deconvolution problem. Importantly, the algorithm progresses through each time series sequentially from beginning to end, thus enabling real-time online spike inference during the imaging session. Our algorithm is a generalization of the pool adjacent violators algorithm (PAVA) for isotonic regression and inherits its linear-time computational complexity. We gain remarkable increases in processing speed: more than one order of magnitude compared to currently employed state of the art convex solvers relying on interior point methods. Our method can exploit warm starts; therefore optimizing model hyperparameters only requires a handful of passes through the data. The algorithm enables real-time simultaneous deconvolution of $O(10^5)$ traces of whole-brain zebrafish imaging data on a laptop.

## 1 Introduction

Calcium imaging has become one of the most widely used techniques for recording activity from neural populations in vivo [1]. The basic principle of calcium imaging is that neural action potentials (or spikes), the point process signal of interest, each induce an optically measurable transient response in calcium dynamics. The nontrivial problem to extract the spike train of each neuron from a raw fluorescence trace has been addressed with several different approaches, including template matching [2] and linear deconvolution [3, 4], which are outperformed by sparse nonnegative deconvolution [5]. The latter can be interpreted as the MAP estimate under a generative model (linear convolution plus noise; Fig. 1), whereas fully Bayesian methods [6, 7] can provide some further improvements, but are more computationally expensive. Supervised methods trained on simultaneously-recorded electrophysiological and imaging data [8, 9] have also recently achieved state of the art results, but are more black-box in nature.

The methods above are typically applied to imaging data offline, after the experiment is complete; however, there is a need for accurate and fast real-time processing to enable closed-loop experiments, a powerful strategy for causal investigation of neural circuitry [10]. In particular, observing and feeding back the effects of circuit interventions on physiologically relevant timescales will be valuable for directly testing whether inferred models of dynamics, connectivity, and causation are accurate in vivo, and recent experimental advances [11, 12] are now enabling work in this direction. Brain-computer interfaces (BCIs) also rely on real-time estimates of neural activity. Whereas most BCI systems rely on electrical recordings, BCIs have been driven by optical signals too [13], providing new insight into how neurons change their activity during learning on a finer spatial scale than possible with intracortical electrodes. Finally, adaptive experimental design approaches [14, 15, 16] also rely on online estimates of neural activity.

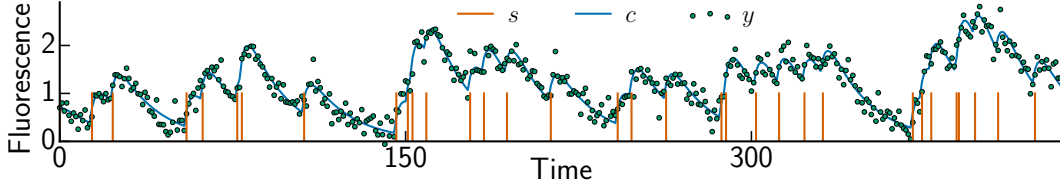

Figure 1: Generative autoregressive model for calcium dynamics. Spike train $s$ gets filtered to produce calcium trace $c$; here we used $p = 2$ as order of the AR process. Added noise yields the observed fluorescence $y$.

Even in cases where we do not require the strict timing/latency constraints of real-time processing, we still need methods that scale to large data sets as for example in whole-brain imaging of larval zebrafish [17, 18]. A further demand for scalability stems from the fact that the deconvolution problem is solved in the inner loop of constrained nonnegative matrix factorization (CNMF) [19], the current state of the art for simultaneous denoising, deconvolution, and demixing of spatiotemporal calcium imaging data.

In this paper we address the pressing need for scalable online spike inference methods. We build on the success of framing spike inference as a sparse nonnegative deconvolution problem. Current algorithms employ interior point methods to solve the ensuing optimization problem and are fast enough to process hundreds of neurons in about the same time as the recording [5], but will not scale to currently obtained larger data sets such as whole-brain zebrafish imaging. Furthermore, these interior point methods scale linearly, but they cannot be warm started, i.e. be initialized with the solution from a previous iteration to gain speed-ups, and do not run online.

We noted a close connection between the MAP problem and isotonic regression, which fits data by a monotone piecewise constant function. A classic isotonic regression algorithm is the pool adjacent violators algorithm (PAVA) [20, 21], which sweeps through the data looking for violations of the monotonicity constraint. When it finds one, it adjusts the estimate to the best possible fit with constraints, which amounts to pooling data points with the same fitted value. During the sweep adjacent pools that violate the constraints are merged. We generalized PAVA to derive an Online Active Set method to Infer Spikes (OASIS) that yields speed-ups in processing time by at least one order of magnitude compared to interior point methods on both simulated and real data. Further, OASIS can be warm-started, which is useful in the inner loop of CNMF, and also when adjusting model hyperparameters, as we show below. Importantly, OASIS is not only much faster, but operates in an online fashion, progressing through the fluorescence time series sequentially from beginning to end. The advances in speed paired with the inherently online fashion of the algorithm enable true real-time online spike inference during the imaging session, with the potential to significantly impact experimental paradigms. We expect our algorithm to be a useful tool for the neuroscience community, to enable new experiments that online access to spike timings affords and to be of interest in other fields, such as physics and quantitative finance, that deal with jump diffusion processes.

The rest of this paper is organized as follows: Section 2 introduces the autoregressive model for calcium dynamics. In Section 3 we derive our active set method for the sparse nonnegative deconvolution problem for the simple case of AR(1) dynamics and generalize it to arbitrary AR($p$) processes in the Supplementary Material. We further use the problem's dual formulation to adjust the sparsity level in a principled way (following [19]), and describe methods for fitting model hyperparameters including the coefficients of the AR process. In Section 4 we show some results on simulated as well as real data. Finally, in Section 5 we conclude with possible further extensions.

## 2 Autoregressive model for calcium dynamics

We assume we observe the fluorescence signal for $T$ timesteps, and denote by $s_t$ the number of spikes that the neuron fired at the $t$-th timestep, $t = 1, ..., T$, cf. Figure 1. We approximate the calcium concentration dynamics $c$ using a stable autoregressive process of order $p$ (AR($p$)) where $p$ is a small positive integer, usually $p = 1$ or 2,

$$c_t = \sum_{i=1}^{p} \gamma_i c_{t-i} + s_t. \tag{1}$$

The observed fluorescence $y \in \mathbb{R}^T$ is related to the calcium concentration as [5, 6, 7]:

$$y_t = a\,c_t + \epsilon_t, \quad \epsilon_t \sim \mathcal{N}(0, \sigma^2) \tag{2}$$

where $a$ is a nonnegative scalar and the noise is assumed to be i.i.d. zero mean Gaussian with variance $\sigma^2$. For the remainder we assume units such that $a = 1$ without loss of generality. The parameters $\gamma_i$ and $\sigma$ can be estimated from the autocovariance function and the power spectral density (PSD) of $\boldsymbol{y}$ respectively [19]. The autocovariance approach assumes that the spiking signal $\boldsymbol{s}$ comes from a homogeneous Poisson process and in practice often gives a crude estimate of $\gamma_i$. We will improve on this below (Fig. 3) by fitting the AR coefficients directly, which leads to better estimates, particularly when the spikes have some significant autocorrelation.

The goal of calcium deconvolution is to extract an estimate of the neural activity $\boldsymbol{s}$ from the vector of observations $\boldsymbol{y}$. As discussed in [5, 19], this leads to the following nonnegative LASSO problem for estimating the calcium concentration:

$$\underset{\boldsymbol{c}}{\text{minimize}} \quad \tfrac{1}{2}\|\boldsymbol{c} - \boldsymbol{y}\|^2 + \lambda\|\boldsymbol{s}\|_1 \quad \text{subject to} \quad \boldsymbol{s} = G\boldsymbol{c} \geq 0 \tag{3}$$

where the $\ell_1$ penalty enforces sparsity of the neural activity and the lower triangular matrix G is defined as:

$$G = \begin{pmatrix} 1 & 0 & 0 & \dots & 0 \\ -\gamma_1 & 1 & 0 & \dots & 0 \\ -\gamma_2 & -\gamma_1 & 1 & \dots & 0 \\ \vdots & \ddots & \ddots & \ddots & \vdots \\ 0 & \dots & -\gamma_2 & -\gamma_1 & 1 \end{pmatrix} \tag{4}$$

Following the approach in [5] the spike signal $\boldsymbol{s}$ is relaxed from nonnegative integers to arbitrary nonnegative values.

# 3 Derivation of the active set algorithm

The optimization problem (3) could be solved using generic convex program solvers. Here we derive the much faster Online Active Set method to Infer Spikes (OASIS).

## 3.1 Online Active Set method to Infer Spikes (OASIS)

For simplicity we consider first the AR(1) model and defer the cumbersome general case $p > 1$ to the Supplementary Material. We begin by inserting the definition of $\boldsymbol{s}$ (Eq. 3, skipping the index of $\gamma$ for a single AR coefficient). Using that $\boldsymbol{s}$ is constrained to be nonnegative yields for the sparsity penalty

$$\lambda\|\boldsymbol{s}\|_1 = \lambda\mathbf{1}^\top\boldsymbol{s} = \lambda\sum_{t=1}^{T}\sum_{k=1}^{T} G_{k,t}c_t = \lambda\sum_{t=1}^{T}(1 - \gamma + \gamma\delta_{tT})c_t = \sum_{t=1}^{T}\mu_t c_t = \boldsymbol{\mu}^\top\boldsymbol{c} \tag{5}$$

with $\mu_t := \lambda(1 - \gamma + \gamma\delta_{tT})$ (with $\delta$ denoting Kronecker's delta) by noting that the sum of the last column of $G$ is 1, whereas all other columns sum to $(1 - \gamma)$. Now the problem

$$\underset{\boldsymbol{c}}{\text{minimize}} \quad \frac{1}{2}\sum_{t=1}^{T}(c_t - y_t)^2 + \sum_{t=1}^{T}\mu_t c_t \quad \text{subject to} \quad c_{t+1} \geq \gamma c_t \geq 0 \quad \forall t \tag{6}$$

shares some similarity to isotonic regression with the constraint $c_{t+1} \geq c_t$. However, our constraint $c_{t+1} \geq \gamma c_t$ bounds the rate of decay instead of enforcing monotonicity. We generalize PAVA to handle the additional factor $\gamma$. The algorithm is based on the following: For an optimal solution, if $y_t < \gamma y_{t-1}$, then the constraint becomes active and holds with equality, $c_t = \gamma c_{t-1}$. (Supposing the opposite, i.e. $c_t > \gamma c_{t-1}$, we could move $c_{t-1}$ and $c_t$ by some small $\epsilon$ to decreases the objective without violating the constraints, yielding a proof by contradiction.)

We first present the algorithm in a way that conveys its core ideas, then improve the algorithm's efficiency by introducing "pools" of variables (adjacent $c_t$ values) which are updated simultaneously. We introduce temporary values $\boldsymbol{c}'$ and initialize them to the unconstrained least squares solution, $\boldsymbol{c}' = \boldsymbol{y} - \boldsymbol{\mu}$. Initially all constraints are in the "passive set" and possible violations are fixed by subsequently adding the respective constraints to the "active set." Starting at $t = 2$ one moves forward until a violation of the constraint $c'_\tau \geq \gamma c'_{\tau-1}$ at some time $\tau$ is detected (Fig. 2A). Now the constraint is added to the active set and enforced by setting $c'_\tau = \gamma c'_{\tau-1}$. Updating the two time steps by minimizing $\frac{1}{2}(y_{\tau-1} - c'_{\tau-1})^2 + \frac{1}{2}(y_\tau - \gamma c'_{\tau-1})^2 + \mu_{\tau-1}c'_{\tau-1} + \mu_\tau\gamma c'_{\tau-1}$ yields an updated value $c'_{\tau-1}$. However, this updated value can violate the constraint $c'_{\tau-1} \geq \gamma c'_{\tau-2}$ and we need to update $c'_{\tau-2}$ as well, etc., until we have backtracked some $\Delta t$ steps to time $\hat{t} = \tau - \Delta t$

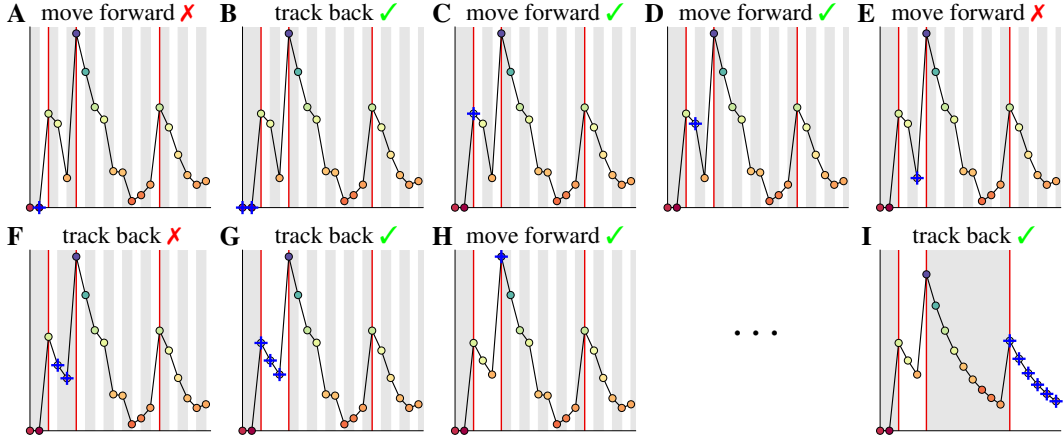

Figure 2: Illustration of OASIS for an AR(1) process (see Supplementary Video). Red lines depict true spike times. The shaded background shows how the time points are gathered in pools. The pool currently under consideration is indicated by the blue crosses. A constraint violation is encountered for the second time step **(A)** leading to backtracking and merging **(B)**. The algorithm proceeds moving forward **(C-E)** until the next violation occurs **(E)** and triggers backtracking and merging **(F-G)** as long as constraints are violated. When the most recent spike time has been reached **(G)** the algorithm proceeds forward again **(H)**. The process continues until the end of the series has been reached **(I)**. The solution is obtained and pools span the inter-spike-intervals.

where the constraint $c'_{\hat{t}} \geq \gamma c'_{\hat{t}-1}$ is already valid. At most one needs to backtrack to the most recent spike, because $c'_{\hat{t}} > \gamma c'_{\hat{t}-1}$ at spike times $\hat{t}$ (Eq. 1). (Because such delays could be too long for some interesting closed loop experiments, we show in the Supplementary Material how well the method performs if backtracking is limited to just few frames.) Solving

$$\underset{c'_{\hat{t}}}{\text{minimize}} \quad \frac{1}{2} \sum_{t=0}^{\Delta t} (\gamma^t c'_{\hat{t}} - y_{t+\hat{t}})^2 + \sum_{t=0}^{\Delta t} \mu_{t+\hat{t}} \gamma^t c'_{\hat{t}} \tag{7}$$

by setting the derivative to zero yields

$$c'_{\hat{t}} = \frac{\sum_{t=0}^{\Delta t} (y_{t+\hat{t}} - \mu_{t+\hat{t}}) \gamma^t}{\sum_{t=0}^{\Delta t} \gamma^{2t}} \tag{8}$$

and the next values are updated according to $c'_{\hat{t}+t} = \gamma^t c'_{\hat{t}}$ for $t = 1, ..., \Delta t$. (Along the way it is worth noting that, because a spike induces a calcium response described by kernel $\boldsymbol{h}$ with components $h_{1+t} = \gamma^t$, $c'_{\hat{t}}$ could be expressed in the more familiar regression form as $\frac{\boldsymbol{h}_{1:\Delta t+1}^\top (\boldsymbol{y}-\boldsymbol{\mu})_{\hat{t}:\tau}}{\boldsymbol{h}_{1:\Delta t+1}^\top \boldsymbol{h}_{1:\Delta t+1}}$, where we used the notation $\boldsymbol{v}_{i:j}$ to describe a vector formed by components $i$ to $j$ of $\boldsymbol{v}$.) Now one moves forward again (Fig. 2C-E) until detection of the next violation (Fig. 2E), backtracks again to the most recent spike (Fig. 2G), etc. Once the end of the time series is reached (Fig. 2I) we have found the optimal solution and set $\boldsymbol{c} = \boldsymbol{c}'$.

In a worst case situation a constraint violation is encountered at every step of the forward sweep through the series. Updating all $t$ values up to time $t$ yields overall $\sum_{t=2}^{T} t = \frac{T(T+1)}{2} - 1$ updates and an $O(T^2)$ algorithm. In order to obtain a more efficient algorithm we introduce pools which are tuples of the form $(v_i, w_i, t_i, l_i)$ with value $v_i$, weight $w_i$, event time $t_i$ and pool length $l_i$. Initially there is a pool $(y_t - \mu_t, 1, t, 1)$ for each time step $t$. During backtracking pools get combined and only the first value $v_i = c'_{t_i}$ is explicitly considered, while the other values are merely defined implicitly via $c_{t+1} = \gamma c_t$. The constraint $c_{t+1} \geq \gamma c_t$ translates to $v_{i+1} \geq \gamma^{l_i} v_i$ as the criterion determining whether pools need to be combined. The introduced weights allow efficient value updates whenever pools are merged by avoiding recalculating the sums in equation (8). Values are updated according to

$$v_i \leftarrow \frac{w_i v_i + \gamma^{l_i} w_{i+1} v_{i+1}}{w_i + \gamma^{2l_i} w_{i+1}} \tag{9}$$

where the denominator is the new weight of the pool and the pool lengths are summed

$$w_i \leftarrow w_i + \gamma^{2l_i} w_{i+1} \tag{10}$$

$$l_i \leftarrow l_i + l_{i+1}. \tag{11}$$

Whenever pools $i$ and $i+1$ are merged, former pool $i+1$ is removed and the succeeding pool indices decreased by 1. It is easy to prove by induction that the updates according to equations (9-11) guarantee that equation (8) holds for all values (see Supplementary Material) without having to explicitly calculate it. The latter would be expensive for long pools, whereas merging two pools has $O(1)$ complexity independent of the pool lengths. With pooling the considered worst case situation results in a single pool that is updated at every step forward, yielding $O(T)$ complexity. Analogous to PAVA, the updates solve equation (6) not just greedily but optimally. The final algorithm is summarized in Algorithm 1 and illustrated in Figure 2 as well as in the Supplementary Video.

---

**Algorithm 1** Fast online deconvolution algorithm for AR(1) processes with positive jumps

---

**Require:** data $y$, decay factor $\gamma$, regularization parameter $\lambda$
1: initialize pools as $\mathcal{P} = \{(v_i, w_i, t_i, l_i)\}_{i=1}^{T} \leftarrow \{(y_t - \lambda(1 - \gamma + \gamma\delta_{tT}), 1, t, 1)\}_{t=1}^{T}$ and let $i \leftarrow 1$
2: **while** $i < |\mathcal{P}|$ **do**                    ▷ iterate until end
3:     **while** $i < |\mathcal{P}|$ and $v_{i+1} \geq \gamma^{l_i} v_i$ **do** $i \leftarrow i+1$        ▷ move forward
4:     **if** $i == |\mathcal{P}|$ **then** break
5:     **while** $i > 0$ and $v_{i+1} < \gamma^{l_i} v_i$ **do**            ▷ track back
6:         $\mathcal{P}_i \leftarrow \left( \frac{w_i v_i + \gamma^{l_i} w_{i+1} v_{i+1}}{w_i + \gamma^{2l_i} w_{i+1}}, w_i + \gamma^{2l_i} w_{i+1}, t_i, l_i + l_{i+1} \right)$        ▷ Eqs. (9-11)
7:         remove $\mathcal{P}_{i+1}$
8:         $i \leftarrow i - 1$
9:     $i \leftarrow i + 1$
10: **for** $(v, w, t, l)$ in $\mathcal{P}$ **do**                ▷ construct solution for all $t$
11:     **for** $\tau = 0, ..., l - 1$ **do** $c_{t+\tau} \leftarrow \gamma^{\tau} \max(0, v)$    ▷ enforce $c_t \geq 0$ via max
12: **return** $c$

---

## 3.2  Dual formulation with hard noise constraint

The formulation above contains a troublesome free sparsity parameter $\lambda$ (implicit in $\boldsymbol{\mu}$). A more robust deconvolution approach eliminates it by inclusion of the residual sum of squares (RSS) as a hard constraint and not as a penalty term in the objective function [19]. The expected RSS satisfies $\langle \|\boldsymbol{c} - \boldsymbol{y}\|^2 \rangle = \sigma^2 T$ and by the law of large numbers $\|\boldsymbol{c} - \boldsymbol{y}\|^2 \approx \sigma^2 T$ with high probability, leading to the constrained problem

$$\underset{\boldsymbol{c}}{\text{minimize}} \quad \|\boldsymbol{s}\|_1 \quad \text{subject to} \quad \boldsymbol{s} = G\boldsymbol{c} \geq 0 \quad \text{and} \quad \|\boldsymbol{c} - \boldsymbol{y}\|^2 \leq \sigma^2 T. \tag{12}$$

(As noted above, we estimate $\sigma$ using the power spectral estimator described in [19].) We will solve this problem by increasing $\lambda$ in the dual formulation until the noise constraint is tight. We start with some small $\lambda$, e.g. $\lambda = 0$, to obtain a first partitioning into pools $\mathcal{P}$, cf. Figure 3A below. From equations (8-10) (and see also S11) along with the definition of $\boldsymbol{\mu}$ (Eq. 5) it follows that given the solution $(v_i, w_i, t_i, l_i)$, where

$$v_i = \frac{\sum_{t=0}^{l_i - 1}(y_{t_i + t} - \mu_{t_i + t})\gamma^t}{\sum_{t=0}^{l_i - 1}\gamma^{2t}} = \frac{\sum_{t=0}^{l_i - 1}(y_{t_i + t} - \lambda(1 - \gamma + \gamma\delta_{t_i + t, T}))\gamma^t}{w_i}$$

for some $\lambda$, the solution $(v_i', w_i', t_i', l_i')$ for $\lambda + \Delta\lambda$ is

$$v_i' = v_i - \Delta\lambda\frac{\sum_{t=0}^{l_i - 1}(1 - \gamma + \gamma\delta_{t_i + t, T})\gamma^t}{w_i} = v_i - \Delta\lambda\frac{1 - \gamma^{l_i}(1 - \delta_{iz})}{w_i} \tag{13}$$

where $z = |\mathcal{P}|$ is the index of the last pool and because pools are updated independently we make the approximation that no changes in the pool structure occur. Inserting equation (13) into the noise constraint (Eq. 12) results in

$$\sum_{i=1}^{z}\sum_{t=0}^{l_i - 1}\left(\left(v_i - \Delta\lambda\frac{1 - \gamma^{l_i}(1 - \delta_{iz})}{w_i}\right)\gamma^t - y_{t_i + t}\right)^2 = \sigma^2 T \tag{14}$$

and solving the quadratic equation yields $\Delta\lambda = \frac{-\beta + \sqrt{\beta^2 - 4\alpha\epsilon}}{2\alpha}$ with $\alpha = \sum_{i,t}\xi_{it}^2$, $\beta = 2\sum_{i,t}\chi_{it}\xi_{it}$ and $\epsilon = \sum_{i,t}\chi_{it}^2 - \sigma^2 T$ where $\xi_{it} = \frac{1 - \gamma^{l_i}(1 - \delta_{iz})}{w_i}\gamma^t$ and $\chi_{it} = y_{t_i + t} - v_i\gamma^t$.

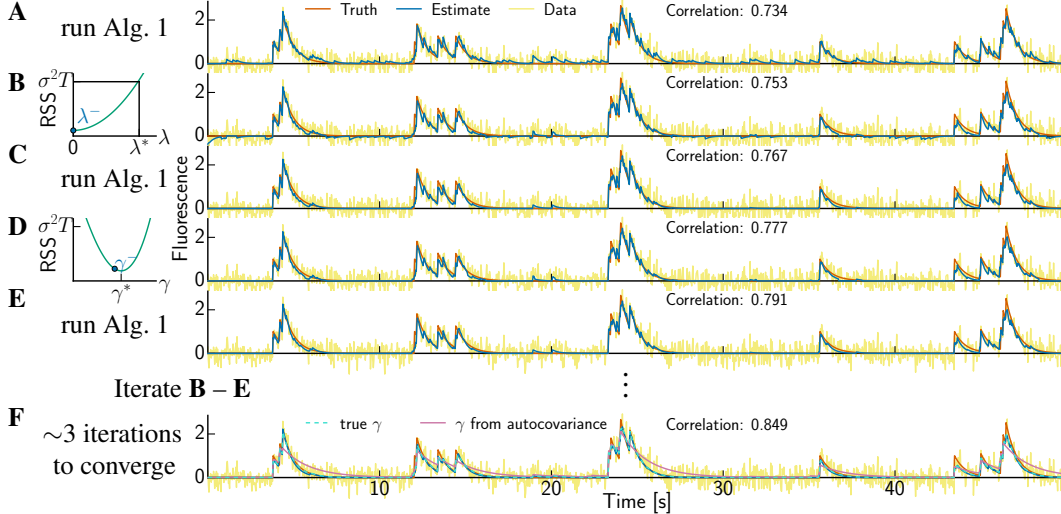

Figure 3: Optimizing sparsity parameter $\lambda$ and AR coefficient $\gamma$. **(A)** Running the active set method, with conservatively small estimate of $\gamma$, yields an initial *denoised* estimate (blue) of the data (yellow) roughly capturing the truth (red). We also report the correlation between the *deconvolved* estimate and true spike train as direct measure for the accuracy of spike train inference. **(B)** Updating sparsity parameter $\lambda$ according to Eq. (14) such that RSS $= \sigma^2 T$ (left) shifts the current estimate downward (right, blue). **(C)** Running the active set method enforces the constraints again and is fast due to warm-starting. **(D)** Updating $\gamma$ by minimizing the polynomial function RSS($\gamma$) and **(E)** running the warm-started active set method completes one iteration, which yields already a decent fit. **(F)** A few more iterations improve the solution further and the obtained estimate is hardly distinguishable from the one obtained with known true $\gamma$ (turquoise dashed on top of blue solid line). Note that determining $\gamma$ based on the autocovariance (purple) yields a crude solution that even misses spikes (at 24.6 s and 46.5 s).

The solution $\Delta\lambda$ provides a good approximate proposal step for updating the pool values $v_i$ (using Eq. 13). Since this update proposal is only approximate it can give rise to violated constraints (e.g., negative values of $v_i$). To satisfy all constraints Algorithm 1 is run to update the pool structure, cf. Figure 3C, but with a *warm start*: we initialize with the current set of merely $z$ pools $\mathcal{P}'$ instead of the $T$ pools for a cold start (Alg. 1, line 1). This step returns a set of $v_i$ values that satisfy the constraints and may merge pools (i.e., delete spikes); then the procedure (update $\lambda$ then rerun the warm-started Algorithm 1) can be iterated until no further pools need to be merged, at which point the procedure has converged. In practice this leads to an increasing sequence of $\lambda$ values (corresponding to an increasingly sparse set of spikes), and no pool-split (i.e., add-spike) moves are necessary[1].

This warm-starting approach brings major speed benefits: after the residual is updated following a $\lambda$ update, the computational cost of the algorithm is linear in the number of pools $z$, hence warm starting drastically reduces computational costs from $k_1 T$ to $k_2 z$ with proportionality constants $k_1$ and $k_2$: if no pool boundary updates are needed then after warm starting the algorithm only needs to pass once through all pools to verify that no constraint is violated, whereas a cold start might involve a couple passes over the data to update pools, so $k_2$ is typically significantly smaller than $k_1$, and $z$ is typically much smaller than $T$ (especially in sparsely-spiking regimes).

### 3.3 Optimizing the AR coefficient

Thus far the parameter $\gamma$ has been known or been estimated based on the autocovariance function. We can improve upon this estimate by optimizing $\gamma$ as well, which is illustrated in Figure 3. After updating $\lambda$ followed by running Algorithm 1, we perform a coordinate descent step in $\gamma$ that minimizes the RSS, cf. Figure 3D. The RSS as a function of $\gamma$ is a high order polynomial, cf. equation (8), and we need to settle for numerical solutions. We used Brent's method [22] with bounds $0 \leq \gamma < 1$. One iteration consists now of steps B-E in Figure 3, while for known $\gamma$ only B-C were necessary.

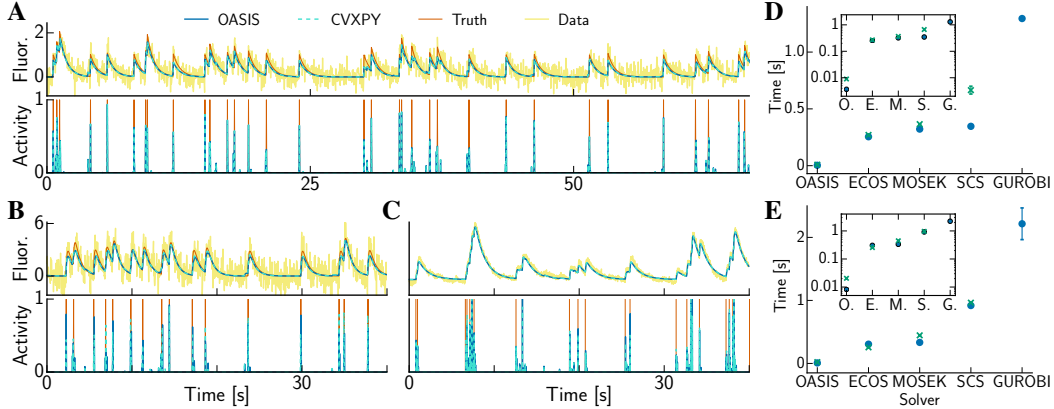

Figure 4: OASIS produces the same high quality results as convex solvers at least an order of magnitude faster. **(A)** Raw and inferred traces for simulated AR(1) data, **(B)** simulated AR(2) and **(C)** real data from [29] modeled as AR(2) process. OASIS solves equation (3) exactly for AR(1) and just approximately for AR(2) processes, nevertheless well extracting spikes. **(D)** Computation time for simulated AR(1) data with given $\lambda$ (blue circles, Eq. 3) or inference with hard noise constraint (green x, Eq. 12). GUROBI failed on the noise constrained problem. **(E)** Computation time for simulated AR(2) data.

## 4 Results

### 4.1 Benchmarking OASIS

We generated datasets of 20 fluorescence traces each for $p = 1$ and 2 with a duration of $100\,\mathrm{s}$ at a framerate of $30\,\mathrm{Hz}$, such that $T = 3,000$ frames. The spiking signal came from a homogeneous Poisson process. We used $\gamma = 0.95$, $\sigma = 0.3$ for the AR(1) model and $\gamma_1 = 1.7$, $\gamma_2 = -0.712$, $\sigma = 1$ for the AR(2) model. Figures 4A-C are reassuring that our suggested (dual) active set method yields indeed the same results as other convex solvers for an AR(1) process and that spikes are extracted well. For an AR(2) process OASIS is greedy and yields good results that are similar to the one obtained with convex solvers (lower panels in Fig. 4B and C), with virtually identical denoised fluorescence traces (upper panels). An exact fast (primal) active set method method for this case is presented in the extended journal version of this paper [23].

Figures 4D,E report the computation time ($\pm$SEM) averaged over all 20 traces and ten runs per trace on a MacBook Pro with Intel Core i5 $2.7\,\mathrm{GHz}$ CPU. We compared the run time of our algorithm to a variety of state of the art convex solvers that can all be conveniently called from the convex optimization toolbox CVXPY [24]: embedded conic solver (ECOS, [25]), MOSEK [26], splitting conic solver (SCS, [27]) and GUROBI [28]. With given sparsity parameter $\lambda$ (Eq. 3) OASIS is about two magnitudes faster than any other method for an AR(1) process (Fig. 4D, blue disks) and more than one magnitude for an AR(2) process (Fig. 4E). Whereas the other solvers take almost the same time for the noise constrained problem (Eq. 12, Fig. 4D,E, green x), our method takes about three times longer to find the value of the dual variable $\lambda$ compared to the formulation where the residual is part of the objective; nevertheless it still outperforms the other algorithms by a huge margin.

We also ran the algorithms on longer traces of length $T = 30,000$ frames, confirming that OASIS scales linearly with $T$. Our active set method maintained its lead by 1-2 orders of magnitude in computing time. Further, compared to our active set method the other algorithms required at least an order of magnitude more RAM, confirming that OASIS is not only faster but much more memory efficient. Indeed, because OASIS can run in online mode the memory footprint can be $O(1)$, instead of $O(T)$.

We verified these results on real data as well. Running OASIS with the hard noise constraint and $p = 2$ on the GCaMP6s dataset collected at $60\,\mathrm{Hz}$ from [29] took $0.101 \pm 0.005\,\mathrm{s}$ per trace, whereas the fastest other methods required $2.37 \pm 0.12\,\mathrm{s}$. Figure 4C shows the real data together with the inferred denoised and deconvolved traces as well as the true spike times, which were obtained by simultaneous electrophysiological recordings [29].

We also extracted each neuron's fluorescence activity using CNMF from an unpublished whole-brain zebrafish imaging dataset from the M. Ahrens lab. Running OASIS with hard noise constraint and

$p = 1$ (chosen because the calcium onset was fast compared to the acquisition rate of 2 Hz) on 10,000 traces out of a total of 91,478 suspected neurons took $81.5\,\mathrm{s}$ whereas ECOS, the fastest competitor, needed $2{,}818.1\,\mathrm{s}$. For all neurons we would hence expect $745\,\mathrm{s}$ for OASIS, which is below the $1{,}500\,\mathrm{s}$ recording duration, and over $25{,}780\,\mathrm{s}$ for ECOS and other candidates.

## 4.2 Hyperparameter optimization

We have shown that we can solve equation (3) and equation (12) faster than interior point methods. The AR coeffient $\gamma$ was either known or estimated based on the autocorrelation in the above analyses. The latter approach assumes that the spiking signal comes from a homogeneous Poisson process, which does not generally hold for realistic data. Therefore we were interested in optimizing not only the sparsity parameter $\lambda$, but also the AR(1) coeffient $\gamma$. To illustrate the optimization of both, we generated a fluorescence trace with spiking signal from an inhomogeneous Poisson process with sinusoidal instantaneous firing rate (Fig. 3), thus mimicking realistic data. We conservatively initialized $\gamma$ to a small value of $0.9$. The value obtained based on the autocorrelation was $0.9792$ and larger than the true value of $0.95$. The left panels in Figures 3B and D illustrate the update of $\lambda$ from the previous value $\lambda^-$ to $\lambda^*$ by solving a quadratic equation analytically (Eq. 14) and the update of $\gamma$ by numerical minimization of a high order polynomial respectively. Note that after merely one iteration (Fig. 3E) a good solution is obtained and after three iterations the solution is virtually identical to the one obtained when the true value of $\gamma$ has been provided (Fig. 3F). This holds not only visually, but also when judged by the correlation between *deconvolved* activity and ground truth spike train, which was $0.869$ compared to merely $0.773$ if $\gamma$ was obtained based on the autocorrelation. The optimization was robust to the initial value of $\gamma$, as long as it was positive and not, or only marginally, greater than the true value. The value obtained based on the autocorrelation was considerably greater and partitioned the time series into pools in a way that missed entire spikes. A quantification of the computing time for hyperparameter optimization as well as means to reduce it are presented in the extended journal version [23].

## 5 Conclusions

We presented an online active set method for spike inference from calcium imaging data. We assumed that the forward model to generate a fluorescence trace from a spike train is linear-Gaussian. Future work will extend the method to nonlinear models [30] incorporating saturation effects and a noise variance that increases with the mean fluorescence to better resemble the Poissonian statistics of photon counts. In the Supplementary Material we already extend our mathematical formulation to include weights for each time point as a first step in this direction.

Further development, contained in the extended journal version [23], includes and optimizes an explicit fluorescence baseline. It also provides means to speed up the optimization of model hyperparameters, including the added baseline. It presents an exact and fast (primal) active set method for AR($p > 1$) processes and more general calcium response kernels. A further extension is to add the constraint that positive spikes need to be larger than some minimal value, which renders the problem non-convex. A minor modification to our algorithm enables it to find an (approximate) solution of this non-convex problem, which can be marginally better than the solution obtained with $\ell_1$ regularizer.

**Acknowledgments**

We would like to thank Misha Ahrens and Yu Mu for providing whole-brain imaging data of larval zebrafish. We thank John Cunningham for fruitful discussions and Scott Linderman as well as Daniel Soudry for valuable comments on the manuscript.

Funding for this research was provided by Swiss National Science Foundation Research Award P300P2_158428, Simons Foundation Global Brain Research Awards 325171 and 365002, ARO MURI W911NF-12-1-0594, NIH BRAIN Initiative R01 EB22913 and R21 EY027592, DARPA N66001- 15-C-4032 (SIMPLEX), and a Google Faculty Research award; in addition, this work was supported by the Intelligence Advanced Research Projects Activity (IARPA) via Department of Interior/ Interior Business Center (DoI/IBC) contract number D16PC00003. The U.S. Government is authorized to reproduce and distribute reprints for Governmental purposes notwithstanding any copyright annotation thereon. Disclaimer: The views and conclusions contained herein are those of the authors and should not be interpreted as necessarily representing the official policies or endorsements, either expressed or implied, of IARPA, DoI/IBC, or the U.S. Government.

## Footnotes

[1]Note that it is possible to cheaply detect any violations of the KKT conditions in a candidate solution; if such a violation is detected, the corresponding pool could be split and the warm-started Algorithm 1 run locally near the detected violations. However, as we noted, due to the increasing $\lambda$ sequence we did not find this step to be necessary in the examples examined here.

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
