[Supplementary Material]

# Fast Active Set Methods for Online Spike Inference from Calcium Imaging – Supplementary Material –

**Johannes Friedrich[1,2], Liam Paninski[1]**
[1]Grossman Center for the Statistics of Mind and Department of Statistics,
Columbia University, New York, NY
[2]Janelia Research Campus, Ashburn, VA
j.friedrich@columbia.edu, liam@stat.columbia.edu

## 1 Generalization beyond the AR(1) case

An AR(1) process models the calcium response to a spike as an instantaneous increase followed by an exponential decay. This is a good description when the fluorescence rise time constant is small compared to the length of a time-bin, e.g. when using GCaMP6f [1] and a low imaging frame rate. For fast imaging rates and slow indicators such as GCaMP6s it is more accurate to explicitly model the finite rise time. Typically we choose an AR(2) process, though more structured responses (e.g. multiple decay time constants) can also be modeled with higher values for the order $p$.

For $p > 1$ the dynamics is not Markovian any more and the next value depends not only on the current but on possibly multiple previous time steps. Due to the non-Markovian dynamics, following along the lines of the AR(1) case, as we do here, just leads to a greedy, approximate solution; we present an exact primal active set algorithm in the extended journal version [2], which is not as fast as the greedy one, but still outperforms interior point methods.

For an AR($p$) process the sparsity penalty $\lambda\|s\|_1$ can again be expressed as $\boldsymbol{\mu}^\top \boldsymbol{c}$, because

$$\lambda\|\boldsymbol{s}\|_1 = \lambda \mathbf{1}^\top \boldsymbol{s} = \lambda \sum_{t=1}^{T}\sum_{k=1}^{T} G_{k,t} c_t = \lambda \sum_{t=1}^{T}(1 - \sum_{i=1}^{\min(p,T-t)} \gamma_i)c_t = \sum_{t=1}^{T} \mu_t c_t = \boldsymbol{\mu}^\top \boldsymbol{c}, \qquad \text{(S1)}$$

with $\mu_t := \lambda(1 - \sum_{i=1}^{\min(p,T-t)} \gamma_i)$, by evaluating the column sums of $G$. We use matrix- and vector notation to describe the dynamics of $c_t$. Let the transition matrix $A$, multi time step calcium vectors $\boldsymbol{\zeta}_t$, and vector $\boldsymbol{e}$ be defined as

$$A = \begin{pmatrix} \gamma_1 & \gamma_2 & \dots & \gamma_p \\ 1 & 0 & \dots & 0 \\ \vdots & \ddots & \ddots & \vdots \\ 0 & \dots & 1 & 0 \end{pmatrix} \qquad \boldsymbol{\zeta}_t = \begin{pmatrix} c_t \\ c_{t-1} \\ \vdots \\ c_{t-p+1} \end{pmatrix} \qquad \boldsymbol{e} = \begin{pmatrix} 1 \\ 0 \\ \vdots \\ 0 \end{pmatrix} \qquad \text{(S2)}$$

The calcium dynamics is given by $\boldsymbol{\zeta}_t = A\boldsymbol{\zeta}_{t-1} + s_t \boldsymbol{e}$. Analogously to the AR(1) case we derive an algorithm that moves through the time series until it finds a violation of the constraint $c'_\tau \geq \boldsymbol{e}^\top A\boldsymbol{\zeta}'_{\tau-1}$ for some time $\tau$, updates $c'_\tau$ and $c'_{\tau-1}$, and backtracks further until the updates do not violate any constraints at previous time steps. Note that we also implicitly have constraints on $\boldsymbol{\zeta}_t$, enforcing the fact that $\boldsymbol{\zeta}_{t+1}$ is mostly a time-shifted version of $\boldsymbol{\zeta}_t$.

Assuming we need to backtrack by $\Delta t$ steps and introducing again $\hat{t} = \tau - \Delta t$, the objective is to minimize $\sum_{t=\hat{t}}^{\tau}(\frac{1}{2}(c'_t - y_t)^2 + \mu_t c'_t)$ with respect to $c'_{\hat{t}}$ under the active constraints $\boldsymbol{\zeta}_t = A\boldsymbol{\zeta}_{t-1}$ for $t = \hat{t}+1, ..., \tau$. Plugging in the constraints on the dynamics the objective reads

$$\underset{c'_{\hat{t}}}{\text{minimize}} \quad \frac{1}{2}\sum_{t=0}^{\Delta t}(\boldsymbol{e}^\top A^t \boldsymbol{\zeta}'_{\hat{t}} - y_{t+\hat{t}})^2 + \sum_{t=0}^{\Delta t} \mu_{t+\hat{t}} \boldsymbol{e}^\top A^t \boldsymbol{\zeta}'_{\hat{t}} \qquad \text{(S3)}$$

Setting the derivative with respect to $c'_{\hat{t}}$ to zero and solving for $c'_{\hat{t}}$ yields

$$c'_{\hat{t}} = \frac{\sum_{t=0}^{\Delta t} \left( y_{t+\hat{t}} - \mu_{t+\hat{t}} - \sum_{k=2}^{p} (A^t)_{1,k} c'_{\hat{t}+1-k} \right) (A^t)_{1,1}}{\sum_{t=0}^{\Delta t} (A^t)_{1,1}^2} \tag{S4}$$

where $(A^t)_{1,1}^2$ denotes the square of the entry in the first row and column in the matrix obtained as $t$-th matrix power of $A$. Again, take note that these entries describe the calcium kernel $\boldsymbol{h}$ with components $h_{1+t} = (A^t)_{1,1}$. Equation (S4) reduces to equation (8) for $p = 1$ where $A$ is just a $1 \times 1$-matrix with entry $\gamma$. The next values are updated according to $c'_{\hat{t}+t} = \sum_{k=1}^{p} \gamma_k c'_{\hat{t}+t-k}$ for $t = 1, ..., \Delta t$.

We derive again an efficient formulation of the algorithm using pools. Considering the denominator in equation (S4) as a weight in analogy to the AR(1) case and calculating the weighted sum upon merging of pools is not valid for $p > 1$ because in general $(A^t)_{1,1}(A^u)_{1,1} \neq (A^{t+u})_{1,1}$. Introducing pools is still useful as it allows us to keep track of only a small number of $p$ elements in each pool. While for the case of AR(1) we only kept track of each group's first element, we now keep track of the first as well as the $p - 1$ last elements. In order to speed up the update in equation (S4), we can precompute the powers of $A$ and store $(A^t)_{1,:}$ in memory. Note that only the powers up to the maximal inter-spike-interval are needed, which can be much smaller than $T$; of course, for very large values of $t$, $(A^t)_{1,:} \approx 0$, by the stability of $A$; thus for high powers the entries of $(A^t)_{1,:}$ can also be well approximated by a quickly computable exponential function or simply be truncated. The possible values of the denominator can be precomputed as well. The final algorithm for AR(2) is summarized in Algorithm S1, with the extension to $p > 2$ being straight forward but tedious. It includes a special treatment of the first pool discussed in the next section.

According to equation (S4) the solution is a linear function of $\boldsymbol{\mu}$, and hence of $\lambda$. Thus the hard noise constraint for the RSS $\|\boldsymbol{c} - \boldsymbol{y}\|^2 = \sigma^2 T$ is a quadratic equation in $\lambda$, that can be solved analytically, under the assumption of invariant pool structure analogous to above case of AR(1), but involving more lengthy expressions which we do not state explicitly. Updating all pools independently according to equation (S4) can give rise to violated constraints, requiring us to rerun the algorithm, warm-started by initializing with the current set of pools, as described in the main text. After 2-3 iterations no pools need to be merged and the final solution has been found. We can again interleave an update step for optimizing the parameters $\gamma_i$, as described in the main text.

## 2 Initial calcium fluorescence

Thus far we have not explicitly taken account of elevated initial calcium fluorescence levels due to previous spiking activity. For the case $p = 1$ positive fluorescence values $c_1$ capture initial calcium fluorescence that decays exponentially. Positive values $c_1$ lead via $\boldsymbol{s} = G\boldsymbol{c}$ to a positive spike $s_1$. Instead of attributing the elevated fluorescence to a spike at time $t = 1$, a positive $s_1$ more likely accounts for previous spiking activity. Therefore we remove the initial spike by setting $s_1 = 0$.

For $p = 2$ we can model the effect of prior spiking activity as an exponential decay, too. Because the validity of the constraint $c_t \geq \sum_{i=1}^{p} \gamma_i c_{t-i}$ can only be evaluated if $t > p$, for $p > 1$ the first pool stays thus far merely at its initialization $(y_1 - \mu_1, y_1 - \mu_1, 1, 1)$, and the noisy raw data value is taken as true $c_1$. Instead, we suggest to use the first pool to model the exponential decay due to previous spiking activity. Given $c_1 = v_1$ the fluorescence values $c_t$ for $t = 1, ..., l_1$ are then given by $d^{t-1} c_1$ with decay variable $d = \frac{1}{2}(\gamma_1 + \sqrt{\gamma_1^2 + 4\gamma_2})$ [3]. The first pool is merged with the second one whenever the constraint $v_2 \geq d^{l_1} v_1$ is violated, cf. Algorithm S1.

## 3 Weighted regression

For sake of generality we consider the case of weighted regression with weights $\boldsymbol{\theta}$.

$$\underset{\boldsymbol{c}}{\text{minimize}} \quad \frac{1}{2} \sum_t \theta_t (c_t - y_t)^2 + \lambda \sum_t s_t \quad \text{subject to} \quad \boldsymbol{s} = G\boldsymbol{c} \geq 0 \tag{S5}$$

These weights could be used to give lower weight to time points with higher variance for heteroscedastic data, for example for the Poissonian statistics of photon counts where the variance of the

**Algorithm S1** Fast greedy online deconvolution algorithm for AR(2) processes with positive jumps

**Require:** data $\boldsymbol{y}$, AR parameters $\gamma_1, \gamma_2$, regularization parameter $\lambda$, upper bound on inter-spike-interval $\mathrm{ISI}_{\max}$

1: **for** $t = 0, ..., \mathrm{ISI}_{\max}$ **do**                                                                                   $\triangleright$ precompute
2:     $(\alpha_t, \beta_t) = (A^t)^\dagger_{1,:}, \ \delta_{t+1} = \sum_{k=0}^t \alpha_k^2, \ \epsilon_{t+1} = \sum_{k=0}^t \alpha_k \beta_k$
3: let $y_t \leftarrow y_t - \lambda \left(1 - \sum_{i=1}^{\min(2,T-t)} \gamma_i \right) \forall t, \ i \leftarrow 1 \ $ and $\ d = \frac{1}{2}(\gamma_1 + \sqrt{\gamma_1^2 + 4\gamma_2})$
4: initialize pools as $\mathcal{P} = \{(v_t, u_t, t_t, l_t)\}_{t=1}^T \leftarrow \{(y_t, y_t, t, 1)\}_{t=1}^T$
5: **while** $i < |\mathcal{P}|$ **do**                                                                                   $\triangleright$ iterate until end
6:     **while** $i < |\mathcal{P}|$ **and** $(v_{i+1} \geq \alpha_{l_i} v_i + \beta_{l_i} u_{i-1}$ **if** $i > 1$ **else** $v_2 \geq du_1)$ **do**          $\triangleright$ move forward
7:         $i \leftarrow i + 1$
8:     **if** $i == |\mathcal{P}|$ **then** break
9:     **while** $i > 0$ **and** $(v_{i+1} < \alpha_{l_i} v_i + \beta_{l_i} u_{i-1}$ **if** $i > 1$ **else** $v_2 < du_1)$ **do**                      $\triangleright$ track back
10:         $l_i \leftarrow l_i + l_{i+1}$
11:         $v_i \leftarrow \dfrac{\sum_{k=0}^{l_i-1} \alpha_k y_{t_i+k} - \epsilon_{l_i} u_{i-1}}{\delta_{l_i}}$ **if** $i > 1$ **else** $\max\left(0, \dfrac{\sum_{t=1}^{l_1} d^{t-1} y_t}{\sum_{t=0}^{l_1-1} d^{2t}}\right)$          $\triangleright$ Eq. (S4,8)
12:         $u_i \leftarrow \alpha_{l_i-1} v_i + \beta_{l_i-1} u_{i-1}$ **if** $i > 1$ **else** $d^{l_1-1} v_1$
13:         remove $\mathcal{P}_{i+1}$
14:         $i \leftarrow i - 1$
15:     $i \leftarrow i + 1$
16: **for** $(v, u, \hat{t}, l)$ in $\mathcal{P}$ **do**                                                                                   $\triangleright$ construct solution for all $t$
17:     $c_{\hat{t}} \leftarrow v$
18:     **for** $t = \hat{t} + 1, ..., \hat{t} + l - 1$ **do**
19:         $c_t \leftarrow \gamma_1 c_{t-1} + \gamma_2 c_{t-2}$ **if** $\hat{t} > 1$ **else** $dc_{t-1}$
20: **return** $\boldsymbol{c}$

---

$\dagger$ The elements of the matrix powers $A^t$ can be efficiently computed using the equivalent expression as difference of two exponentials well known in the AR / linear systems literature [3]: $(A^t)_{1,1} = \frac{d^{t+1} - r^{t+1}}{d-r}$ and $(A^t)_{1,2} = \gamma_2 \frac{d^t - r^t}{d-r}$, with decay variable $d = \frac{1}{2}(\gamma_1 + \sqrt{\gamma_1^2 + 4\gamma_2})$ and rise variable $r = \frac{1}{2}(\gamma_1 - \sqrt{\gamma_1^2 + 4\gamma_2})$.

fluorescence increases with its mean. Further, instead of the linear relationship between fluorescence and calcium concentration (Eq. 2) we could have a nonlinear observation model

$$y_t = f(c_t) + \epsilon_t \tag{S6}$$

where the nonlinear function $f$ can include saturation effects. It is often taken to be the Hill equation, i.e., $f(c) = \frac{ac^n}{c^n + k_d} + b$, with Hill coefficient $n$, dissociation constant $k_d$, scaling factor $a$ and baseline $b$ [4]. Applying Newton's algorithm to optimize for $\boldsymbol{s}$ (or equivalently $\boldsymbol{c}$) results for each Newton step in a weighted constrained regression problem as in equation (S5).

For an AR(1) process introducing weights changes equation (7) to

$$\underset{c'_{\hat{t}}}{\text{minimize}} \quad \frac{1}{2} \sum_{t=0}^{\Delta t} \theta_{t+\hat{t}} (\gamma^t c'_{\hat{t}} - y_{t+\hat{t}})^2 + \sum_{t=0}^{\Delta t} \mu_{t+\hat{t}} \gamma^t c'_{\hat{t}} \tag{S7}$$

and its solution is a modification of equation (8) by adding the weights

$$c'_{\hat{t}} = \frac{\sum_{t=0}^{\Delta t} (\theta_{t+\hat{t}} y_{t+\hat{t}} - \mu_{t+\hat{t}}) \gamma^t}{\sum_{t=0}^{\Delta t} \theta_{t+\hat{t}} \gamma^{2t}} \tag{S8}$$

We merely need to initialize each pool as $(v_t, w_t, t_t, l_t) = (y_t - \frac{\mu_t}{\theta_t}, \theta_t, t, 1)$ for each time step $t$ and the updates according to equations (9-11) guarantee that equation (S8) holds for all values $v_i = c'_{t_i}$ as we prove in the next section.

For an AR($p$) process introducing weights changes equation (S4) to

$$c'_{\hat{t}} = \frac{\sum_{t=0}^{\Delta t} \left( \theta_{t+\hat{t}} \left( y_{t+\hat{t}} - \sum_{k=2}^p (A^t)_{1,k} c'_{\hat{t}+1-k} \right) - \mu_{t+\hat{t}} \right) (A^t)_{1,1}}{\sum_{t=0}^{\Delta t} \theta_{t+\hat{t}} (A^t)_{1,1}^2} \tag{S9}$$

and the same modified initialization holds.

## 4  Validity of updates according to equations (9-11)

**Theorem 1.** *The updates according to equations (9-11) guarantee that equations (8, S8) hold for all values $v_i = c'_{t_i}$.*

*Proof.* We will prove above theorem by induction.

**Assumption:** Let for the denominator and numerator of equation (S8) hold

$$w_i = \sum_{t=0}^{l_i-1} \theta_{t+t_i} \gamma^{2t} \tag{S10}$$

and

$$w_i v_i = \sum_{t=0}^{l_i-1} \left( \theta_{t+t_i} y_{t+t_i} - \mu_{t+t_i} \right) \gamma^{t} \tag{S11}$$

**Base case:** Pools are initialized as $(v_t, w_t, t_t, l_t) = (y_t - \frac{\mu_t}{\theta_t}, \theta_t, t, 1)$ for all $t$, such that equations (S10, S11) hold.

**Induction step:** Consider two pools $(v_i, w_i, t_i, l_i)$ and $(v_{i+1}, w_{i+1}, t_{i+1}, l_{i+1})$ that satisfy equations (S10, S11) and are merged to pool $(v'_i, w'_i, t'_i, l'_i)$ according to equations (9-11).

$$w'_i = w_i + \gamma^{2l_i} w_{i+1} = \sum_{t=0}^{l_i-1} \theta_{t+t_i} \gamma^{2t} + \sum_{t=0}^{l_{i+1}-1} \theta_{t+t_{i+1}} \gamma^{2l_i} \gamma^{2t}$$

$$= \sum_{t=0}^{l_i+l_{i+1}-1} \theta_{t+t_i} \gamma^{2t} = \sum_{t=0}^{l'_i-1} \theta_{t+t'_i} \gamma^{2t}$$

where we used $t_{i+1} = t_i + l_i$. Thus after the update equation (S10) holds for the merged pool too. Now it just remains to show this also for the values:

$$w'_i v'_i = w_i v_i + \gamma^{l_i} w_{i+1} v_{i+1}$$

$$= \sum_{t=0}^{l_i-1} \left( \theta_{t+t_i} y_{t+t_i} - \mu_{t+t_i} \right) \gamma^{t} + \sum_{t=0}^{l_{i+1}-1} \left( \theta_{t+t_{i+1}} y_{t+t_{i+1}} - \mu_{t+t_{i+1}} \right) \gamma^{l_i} \gamma^{t}$$

$$= \sum_{t=0}^{l_i+l_{i+1}-1} \left( \theta_{t+t_i} y_{t+t_i} - \mu_{t+t_i} \right) \gamma^{t} = \sum_{t=0}^{l'_i-1} \left( \theta_{t+t'_i} y_{t+t'_i} - \mu_{t+t'_i} \right) \gamma^{t}$$

$\square$

## 5  Online spike inference with limited lag

For an exact solution of the nonnegative deconvolution problem of an AR(1) process OASIS needs to backtrack to the most recent spike. Such delays could be too long for some interesting closed loop experiments; therefore we were interested in how well the method performs if backtracking is limited to just few frames. We varied the lag in the online estimator, i.e. the number of future samples observed before assigning a spike at time zero, for different signal-to-noise ratios (SNR). For each lag we chose the sparsity parameter $\lambda$ such that the noise constraint $\|c - y\|^2 \leq \sigma^2 T$ was tight. This yielded increasing values of $\lambda$ for smaller lags, compensating for the fact that limiting backtracking to fewer frames also imposes fewer constraints ($c_t \geq \gamma c_{t-1}$) on the dynamics.

The obtained results are depicted in Figure S1. For realistic SNR (3-5, though [1] report even higher values, cf. Fig. 4C) and sample rates (30 Hz), lags of 3-5 yielded similar results as offline estimation. The exact number depends on the noise; however, the main effect of noise was to reduce the optimal performance attainable even with batch processing, as the asymptotic values in Figure S1A reveal. In the extended journal version [2] we show how the results further improve by introducing a positive threshold $s_{\min}$ on the spike size.

Figure S1: Varied lag in the online estimator. (A) Performance of spike inference as function of lag for various noise levels (i.e., inverse SNR). We used correlation of the inferred spike train as similarity measure and compared to ground truth as well as to the optimally recoverable activity when the lag is unlimited as in offline processing. (B) Inferred trace for increasing lag using the data depicted in Fig. 4A with high noise level ($\sigma = 0.3$). The gray lines indicate the true spike times.

# 6 Supplementary video

The supplementary video illustrates OASIS for an AR(1) process. As in Figure 2, red lines depict true spike times and the shaded background shows how the time points are gathered in pools. The pool currently under consideration is indicated by the blue crosses. The upper panel shows how the calcium fluorescence trace $c'$ develops while the algorithm runs, cf. Figure 2. The video additionally shows the deconvolved trace $s' = Gc'$ (Eq. 3) in the lower panel. The algorithm sweeps through the time series and enforces the constraint $s' \geq 0$.

# 7 Code availability

We provide a Python implementation of our algorithm online (https://github.com/j-friedrich/OASIS). The code is readily usable on new data and includes example scripts that produce all figures of this paper.