[Reviews · NeurIPS 2016]

Reviewer 1

Summary

This paper describes an online optimization algorithm for the non-negative spike deconvolution model of Vogelstein et al and Pnevmatikakis et al, inspired by classical algorithms known as PAVA. A proof is shown and the speed of the algorithm is demonstrated on real data.

Qualitative Assessment

Following the authors' rebuttal I maintain my scores. I do agree this is a good algorithm to have for spike deconvolution, but I would have liked the closed loop scenario more convincingly argued for: does online deconvolution actually improve SNR over raw fluorescence, if you were to decode from it in a BMI setting, for example? While the method appears robust and fast, I have some reservation as to the motivation for the algorithm. First, there are other very fast deconvolution methods out there, like the supervised algorithms of Ref 8,9. It is not true (as claimed, line 28) that these algorithms require paired recordings: Theis et al show that their classifiers readily generalize across datasets, and offer improved performance compared to the Paninski methods. Second, I don’t see a “pressing need” (line 59) for an online spike deconvolution algorithm for closed loop experiments. Certainly most researchers do not use spike deconvolution at all and prefer to use the raw fluorescence traces, which are less processed and more “trustworthy”. In an online setting, waiting to see many samples before assigning spikes would not allow for very interesting closed loop experiments. Arguably, even delays of 100ms are too long for BCI interfaces. The authors would need to demonstrate such short delays with their methods, and the SNR advantage it might bring over the raw fluorescence, if they want to argue that their method will be useful for such experiments. Adding such an analysis would significantly improve the relevance of this paper.

Confidence in this Review

2-Confident (read it all; understood it all reasonably well)


Reviewer 2

Summary

This paper presents a fast online active set method to solve the sparse nonnegative deconvolution problem for spike inference. The author provides a special active set algorithm that can progress through each time series sequentially from beginning to the end. The speed of the method is competitive against some state-of-art convex solvers, and enables real-time simultaneous deconvolution. A numerical experiment on whole-brain zebra-fish data is given to illustrate the performance of the new method over existing algorithms.

Qualitative Assessment

This paper proposes a sparse model for spike inference from calcium imaging data. The model seems to be new, but the setting (3) has been studied in different fields (imaging science, statistical learning, etc) and there exist many methods for solving this problem. But due to the nonnegativity of Gc, this problem can be simply written as a special quadratic program. The authors propose an active set method to solve this problem, the generalized weighted regression AR (1), and process and following AR (p) process which makes the theory more complete and rigorous. The validity of updates (proof) is based on induction and gradual adjustment, which is typical among the field. The main innovation of this algorithm is that it can provide scalable online inference, while the current interior point methods may not run online due to the lack of warm-start. This is also clearly shown in the supplementary video, which gives a strong evidence to indicate the goodness of the method. The downside of such a scheme is that the author did not give a rule of choosing the decay factor and regularization parameter, and the worst case (splitting) in theory in 4.1 part does not occur hence I don’t know how it will perform if it really happens.

Confidence in this Review

1-Less confident (might not have understood significant parts)


Reviewer 3

Summary

This paper introduces a new algorithm for inferring the spiking pattern from a univariate calcium-imaging time-series. Calcium imaging is a popular technique for recording activity from thousands of individual neurons simultaneously. An neuronal spike causes a sharp increase but a slow decrease in the observed calcium image signal. Most neuroscience applications require deconvolving the observed signal to recover the underlying spikes. This paper follows quite closely the exponential-decay and noise model [5], introducing an active set algorithm that solves this model much faster than previous methods (they report a 20x difference from previous competitor). They show the speed gain (with apparent successful deconvolution) on both simulated and real data, though it is hard to tell if the quality of spike inference has been improved or degraded.

Qualitative Assessment

I enjoyed reading this paper, as it provides a clear and well argued algorithmic alternative for the spike inference problem. The authors take a different approach than previous work solving the model for this problem, by modifying the isotonic regression into a locally constrained estimation problem. The authors make a strong case that algorithmic acceleration could potentially broaden the experimental scenarios in which large-scale calcium imaging can be used. Main Comments: 1. The paper really only focuses on fitting the non-negative LASSO problem described in Eq (3) [l 105] or on providing the best reconstruction of the path [see how accuracy is measured in Figure 3]. The reconstruction is never directly measured in terms of the spikes location or strength. It is telling that the "goal of calcium deconvolution" is defined in terms of this penalised optimisation, rather than in terms of the underlying spikes. These are related goals, but are not exactly the same. Note that reconstruction equivalency to other algorithms is taken for granted. In particular, its unclear that the hard noise constraint in Part 4 induces better recovery of the spikes compared to other values. A different metric than signal correlation should be used to estimate this. 2. How sensitive is the hard-constraint of part-4 to the exactness of the model. The mean shape in the model (an exponential decay) is quite restrictive; what happens if the model mildly deviates from it? Is the noise still well estimated? Minor comment: The model in Eq 2, if based on previous literature should be referenced [is it from Ref 5?].

Confidence in this Review

2-Confident (read it all; understood it all reasonably well)


Reviewer 4

Summary

This paper proposes a fast online active set method (OASIS) for sparse nonnegative deconvolution problem for spike inference. The authors elaborated solving AR(p) process in detail which is the key part of OASIS. The author also discusses the corresponding dual formulation with hard noise constraints. Experiments are conducted to show the efficiency of the proposed methods.

Qualitative Assessment

1. In section 3, “\|s\|_1=1^TGc is proportional to \|c\|_1”seems non-trivial. Could the authors elaborate more? 2. To help better understand the motivation, the authors may consider introducing the isotonic regression first in section 2. 3. It seems that problem (3) can be solved by ADMM. Could the authors explain the advantages of OASIS over ADMM?

Confidence in this Review

1-Less confident (might not have understood significant parts)


Reviewer 5

Summary

This paper proposes an online active-set method, OASIS, for detection of neural spiking from calcium fluorescence imaging. The algorithm is based on the theory of AR(p) processes. The algorithm performs as well as out-of-the-box methods for fitting AR models, but 10X faster and with a potentially O(1) memory footprint.

Qualitative Assessment

The reported speed-up of CA-imaging analysis by the proposed algorithm is extremely interesting. However, the paper is at many points opaquely written and unnecessarily hard and tedious to follow. Specific comments: 1) ”Real data" in Fig 4C is l looks almost unrealistically clean, not like the typical data from an experiment. But the noise in simulated data looks more typical. 2) A direct performance comparison to Bayesian methods or linear deconvolution would be important to include. Will almost surely be worse (but slower) than former but better than latter, but a quantitative comparison would be helpful for an experimentalist when choosing between methods. 3) Page 4: Add better explanation of the idea of introducing the pools. 4) Page 8: Give all run times in seconds.

Confidence in this Review

2-Confident (read it all; understood it all reasonably well)